# Hyperbolic nodal band structures and knot invariants

**Marcus Stålhammar[1,★], Lukas Rødland[2], Gregory Arone[3],
Jan Carl Budich[4] and Emil J. Bergholtz[1]**

**1** Department of Physics, Stockholm University,
AlbaNova University Center, 106 91 Stockholm, Sweden
**2** University of Oslo, P.B. 1161 Blindern, 0318 Oslo, Norway
**3** Department of Mathematics, Stockholm University, Kräftriket, 106 91 Stockholm, Sweden
**4** Institute of Theoretical Physics, Technische Universität Dresden, 01062 Dresden, Germany

★ marcus.stalhammar@fysik.su.se

## Abstract

We extend the list of known band structure topologies to include a large family of hyperbolic nodal links and knots, occurring both in conventional Hermitian systems where their stability relies on discrete symmetries, and in the dissipative non-Hermitian realm where the knotted nodal lines are generic and thus stable towards any small perturbation. We show that these nodal structures, taking the forms of Turk's head knots, appear in both continuum- and lattice models with relatively short-ranged hopping that is within experimental reach. To determine the topology of the nodal structures, we devise an efficient algorithm for computing the Alexander polynomial, linking numbers and higher order Milnor invariants based on an approximate and well controlled parameterisation of the knot.

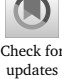
# 1  Introduction

Topology, the theory of global properties that are invariant under continuous deformation, has become a crucial asset for understanding the variety of forms of matter occurring in nature. In particular, the classification of crystalline solids has recently been revolutionised by the advent of topological insulators [1, 2] and semimetals [3] which are distinguished by topological invariants characterising their Bloch bands [4]. Another paradigmatic example for the deep relation and mutual stimulation of topology and physics is provided by knot theory, where the pioneering work by Witten [5] has led to a clear quantum mechanical explanation of the celebrated Jones Polynomial from mathematical knot theory in the physical framework of topological quantum field theory.

Very recently, the two concepts of knots and topological properties of energy bands have been brought together with the discovery of three-dimensional (3D) semimetals that exhibit nodal lines in the form of topologically nontrivial knots [6–11]. In conventional Hermitian systems, the very appearance of nodal lines relies on symmetries, as band touchings there are generically stable only at isolated points in reciprocal space [12]. Quite remarkably, non-Hermitian terms in the Hamiltonian, accounting for the dissipative coupling of a system to its environment, can promote the phenomenon of knotted nodal lines to a topologically robust phenomenon that does no longer rely on any symmetries [13–19].

Motivated by these developments, in this work we contribute to a comprehensive understanding of knotted nodal materials in a twofold way: First, we systematically construct microscopic tight-binding models that feature nodal lines the shapes of which cover a large family of hyperbolic links and knots. Second, we address the question of how to constructively compute (not determine by visual inspection) knot invariants such as the Alexander polynomial and higher order Milnor invariants for any given model Hamiltonian. To this end, an approximate, i.e. topology preserving, analytical parameterisation of the nodal lines is obtained in a variational fashion, based on which knot invariants can be explicitly determined. This contrasts the previous works on nodal links and knots [6–11, 13–19], which focus exclusively on simpler (non-hyperbolic) knots and links, and, crucially, do not provide an algorithmic method for classifying the linking and knottedness of the nodal band structures.

# 2  Microscopic Models for Knotted Nodal Lines

A non-interacting particle-hole symmetric two-band system is completely described in reciprocal space by a Bloch Hamiltonian

$$H(\mathbf{k}) = \mathbf{d}(\mathbf{k}) \cdot \boldsymbol{\sigma} + d_0(\mathbf{k})\sigma_0, \tag{1}$$

where $\mathbf{d}(\mathbf{k}) = (d_x(\mathbf{k}), d_y(\mathbf{k}), d_z(\mathbf{k})) \in \mathbb{R}^3$, $d_0(\mathbf{k}) \in \mathbb{R}$, $\boldsymbol{\sigma}$ is the vector of Pauli matrices, $\sigma_0$ is the $2 \times 2$ identity matrix and $\mathbf{k}$ the lattice momentum with components $k_x$, $k_y$ and $k_z$. Note

that we restrict ourselves to 3D models. By performing a Fourier transformation a real space Hamiltonian can be obtained. Each harmonic appearing in the reciprocal space Hamiltonian Eq. (1) is then directly related to the respective hopping distance in the real space model, i.e., the hopping distances in a real space description are explicitly given by the degree of the corresponding momentum components in the reciprocal model. The eigenvalues of $H(\mathbf{k})$ read as

$$E_{\pm} = d_0 \pm \sqrt{d_x^2 + d_y^2 + d_z^2}. \tag{2}$$

In this work, we are interested in band intersections, which are given by $E_+ = E_-$, hence we can without loss of generality set $d_0 = 0$. Band intersections in Hermitian systems, such as Eq. (1), are in 3D generically stable at points, which is easily seen by counting parameters [12]. Explicitly the intersections are given by

$$d_x^2 + d_y^2 + d_z^2 = 0, \tag{3}$$

resulting in that $d_x = d_y = d_z = 0$ is the only possibility. This yields a system of equations with three equations and three variables, $k_x$, $k_y$ and $k_z$, giving point-like solutions called nodal points. Higher dimensional solutions can however appear, if for example two of the three equations are equivalent. Such a fine tuned system is generically unstable, and the resulting nodal line will dissolve into points as soon as the system is perturbed. The presence of symmetries in a system can however increase the dimension of the nodal structure which is then stable to symmetry preserving perturbations. If, e.g. PT-symmetry is imposed, the Hamiltonian must be real, meaning that $d_y(\mathbf{k})$ has to vanish for all $\mathbf{k}$. A PT-symmetric Hamiltonian thus attains the form,

$$H_{\mathrm{PT}} = d_x(\mathbf{k})\sigma_x + d_z(\mathbf{k})\sigma_z \tag{4}$$

and the nodal structure is given by $d_x^2 + d_z^2 = 0$. Parameter counting tells us that the common solutions of the two equations $d_x = 0$ and $d_z = 0$ form lines. These systems are stable in the sense that sufficiently small symmetry preserving perturbations do not dissolve the nodal lines into points. The nodal lines can attain many different shapes, from open lines to closed curves which can be linked [6–9], or even knotted [11].

In recent years, topological aspects of dissipative systems where e.g. gain and loss are accounted for, have been studied extensively [20–23]. Such systems can be described by adding an imaginary component to the $\mathbf{d}$-vector, resulting in a non-Hermitian Hamiltonian, written in Bloch form as

$$H(\mathbf{k}) = \mathbf{d}_R(\mathbf{k}) \cdot \boldsymbol{\sigma} + i\mathbf{d}_I(\mathbf{k}) \cdot \boldsymbol{\sigma}, \tag{5}$$

where $\mathbf{d}_R(\mathbf{k}), \mathbf{d}_I(\mathbf{k}) \in \mathbb{R}^3$ and the term of the form $d_0\sigma_0$ is disregarded as argued above. The squared eigenvalues of such an operator reads

$$E^2(\mathbf{k}) = \mathbf{d}_R^2(\mathbf{k}) - \mathbf{d}_I^2(\mathbf{k}) + 2i\mathbf{d}_R(\mathbf{k}) \cdot \mathbf{d}_I(\mathbf{k}) \tag{6}$$

and thus the nodal points are given by,

$$\mathrm{Re}[E^2] = \mathbf{d}_R^2 - \mathbf{d}_I^2 = 0, \quad \mathrm{Im}[E^2] = 2\mathbf{d}_R \cdot \mathbf{d}_I = 0. \tag{7}$$

By again counting the parameters, we see that the solutions of Eq. (7) in three dimensions generically occur as 1D contours in reciprocal space that are closed for crystalline systems. Generally, they appear at finite $\mathbf{d} = \mathbf{d}_R + \mathbf{d}_I$, and thus the nodal lines consist of exceptional points, except for when $\mathbf{d} = \mathbf{0}$ [13, 24]. Such line solutions are, in contrast to any Hermitian counterpart, generically stable towards any small perturbation and therefore do not rely on any fine tuning or symmetries. As in the Hermitian case, the exceptional lines can be linked [14–16] or knotted [18, 19]. What is common for nodal lines in Hermitian and non-Hermitian systems is that they appear as the intersection between two implicitly defined 2D surfaces.

## 2.1 Construction of Hamiltonians with a Desired Nodal Topology

Let $s_1$ and $s_2$ be two 2D surfaces in momentum space, whose intersection $s_1 \cap s_2$ is 1D. Assume that there are two continuously differentiable functions $f_1$ and $f_2$ that vanish exactly on $s_1$ and $s_2$ respectively. Then, the topology of the intersection can be retrieved as exceptional structures in a 3D non-Hermitian Hamitlonian by explicit construction of a **d**-vector, e.g. as [18],

$$\mathbf{d}_R = (f_1 - \Lambda, \Lambda, 0), \quad \mathbf{d}_I = (0, f_2, \sqrt{2}\Lambda), \tag{8}$$

where $\Lambda$ is a real, but non-zero, constant. The exceptional points of such a system are given by,

$$\mathrm{Re}[E^2] = f_1^2 - f_2^2 - 2f_1\Lambda = 0, \quad \mathrm{Im}[E^2] = 2f_2\Lambda = 0, \tag{9}$$

which is fulfilled when $f_2 = 0$ and $f_1(f_1 - 2\Lambda) = 0$. In the limit $|\Lambda| \to \infty$, the solutions become $f_1 = f_2 = 0$, which exactly agrees with $s_1 \cap s_2$. For finite, but sufficiently large $|\Lambda|$, these solutions are deformed, but they still inherit the topology of $s_1 \cap s_2$.

Conventional Hermitian models with topologically equivalent nodal structures can be constructed in a similar way. For example, a dual Hermitian model can be obtained from the non-Hermitian model given by Eq. (8), e.g. as [25],

$$\mathbf{d}_{\mathrm{dual}} = (1, 0, 0)(\mathbf{d}_R^2 - \mathbf{d}_I^2) + (0, 1, 0)(\mathbf{d}_R \cdot \mathbf{d}_I). \tag{10}$$

The corresponding nodal structures coincides exactly with that of the non-Hermitian model. Note that this dual model will have twice as large hopping distance as the corresponding non-Hermitian description, since it is quadratic in $f_1$ and $f_2$. By exploiting symmetries, this can be avoided, and more importantly it provides insights into the requirements from stabilising the nodal topology of the band structure. Saliently, the PT-symmetric model given by

$$\mathbf{d}_{\mathrm{PT}} = (f_1, 0, f_2), \tag{11}$$

hosts nodal structures given by $f_1 = f_2 = 0$, which coincides exactly with $s_1 \cap s_2$. Since this model is only linear in $f_1$ and $f_2$, the hopping distance is halved compared to the dual Hermitian model Eq. (11).

An important difference between Hermitian and non-Hermitian models is, however, given by the behaviour under finite perturbations. By adding any perturbation to Eq. (10), the nodal topology collapses to point singularities. The same is true if an arbitrary small symmetry breaking term is added to Eq. (11), but as long as the perturbation preserves the pertinent PT symmetry, the topology is maintained for sufficiently small perturbations. By contrast to these Hermitian cases, the exceptional lines occurring in non-Hermitian systems are generically stable towards any sufficiently small perturbation, and thus they do not rely on fine tuning or symmetries.

The above reasoning has resulted in the discovery of nodal torus knots, both in the Hermitian [11] and non-Hermitian realm [18]. Recall that for a pair of coprime integers $p$ and $q$, the $(p,q)$-torus knot is described by the zeros of a complex polynomial $\xi^p + \zeta^q$ lying on $S_\epsilon^3$, the three-sphere with small radius $\epsilon$ [26]. Defining $f_1 = \mathrm{Re}[\xi^p + \zeta^q]$ and $f_2 = \mathrm{Im}[\xi^p + \zeta^q]$ constrained on $S_\epsilon^3$, mapping them to the Brillouin zone and using the above described **d**-vectors results in nodal torus knots. If $p$ and $q$ are not coprime, the zeros of $\xi^p + \zeta^q$ on $S_\epsilon^3$ represent torus links with $n = \mathrm{GCD}(p, q)$ components of $(\frac{p}{n}, \frac{q}{n})$-torus knots.

## 2.2 Hyperbolic Nodal Structures

The world of knots and links goes beyond the family of torus links, and as long as a link can be described by intersections of surfaces, they can readily be realised as nodal structures. In
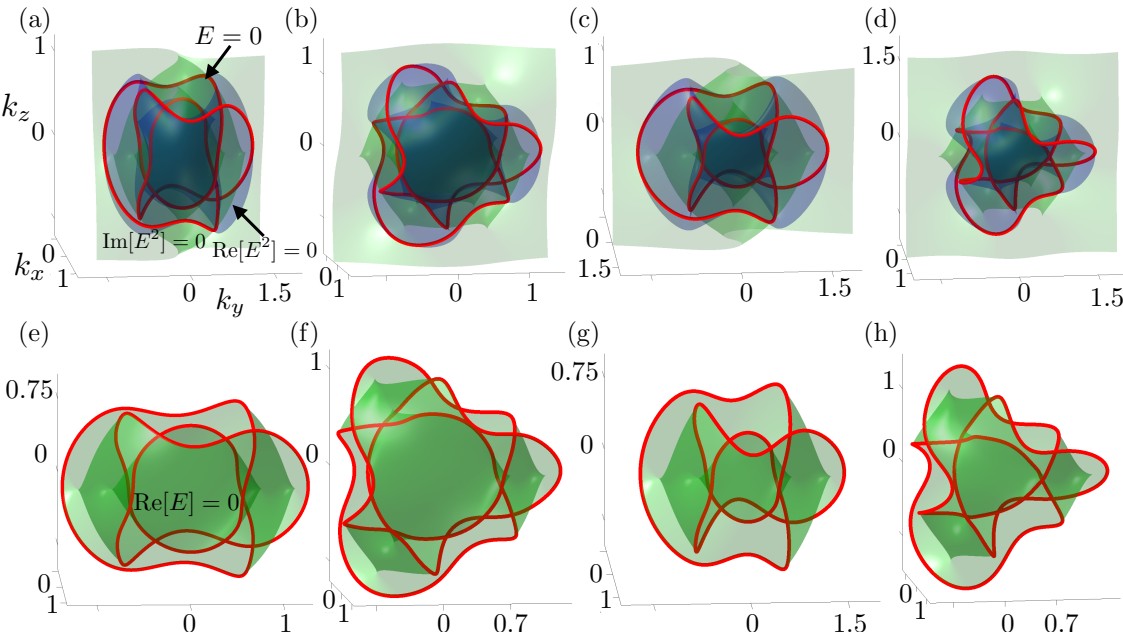

Figure 1: Illustration of hyperbolic exceptional structures, formed as the intersection between the two implicitly defined surfaces $\text{Re}[E^2] = 0$ and $\text{Im}[E^2] = 0$. The upper panel (a)-(d) shows this intersection explicitly, where (a) and (b) are continuum models with $\epsilon^2 = 1$ and $\Lambda = 100$, and (c) and (d) are lattice models with $\epsilon^2 = 0.5$ and $\Lambda = 100$. The lower panel (e)-(h) shows the exceptional structures along with their corresponding Fermi surface. (a) and (c) are figure-eight knots in continuum and lattice models respectively, while (b) and (d) are Borromean rings. The topologies of the nodal structures are determined by calculating the Alexander polynomial for all components. In the case of links, the linking number and, if necessary, higher-order Milnor invariants are also calculated.

Ref. [27], it was explicitly shown that the figure-eight knot and the Borromean rings can be described as the zeros of a real polynomial, constrained to $S^3_\epsilon$. The polynomial is explicitly given by

$$F\left(x, y, \text{Re}[(z + it)^n], \text{Im}[(z + it)^n]\right), \tag{12}$$

where $n$ is an integer, and

$$
\begin{aligned}
F(x, y, z, t) = & \left[(z(x^2 + y^2 + z^2 + t^2) + x\left(8x^2 - 2(x^2 + y^2 + z^2 + t^2)\right)\right. \\
& \left. \sqrt{2}tx + y\left(8x^2 - (x^2 + y^2 + z^2 + t^2)\right)\right],
\end{aligned}
\tag{13}
$$

where $x, y, z, t$ are coordinates on $\mathbb{R}^4$. The zeros of Eq. (12) on $S^3_\epsilon$ can be encoded as nodal structures of a Hamiltonian in the following steps

1. Define $f_1$ and $f_2$ such that their common zeros correspond to the zeros of $F\left(x, y, \text{Re}[(z + it)^n], \text{Im}[(z + it)^n]\right)$.

2. Restrict the zeros to $S^3_\epsilon$.

3. Introduce the lattice momentum by performing the desired change of coordinates.

Following these steps, we explicitly construct models hosting hyperbolic knots and links. First, we treat two specific examples where $n = 2$ and $n = 3$, yielding a nodal figure-eight knot and Borromean rings respectively, with further more exotic examples obtained for higher $n$

discussed later. The functions $f_1$ and $f_2$ are introduced as the different components of $F$ respectively,

$$f_1^{\text{F8}} = (z^2 - t^2)\epsilon^2 + x(8x^2 - 2\epsilon^2), \qquad f_2^{\text{F8}} = 2\sqrt{2}xzt + y(8x^2 - \epsilon^2), \qquad (14)$$

$$f_1^{\text{BR}} = (z^3 - 3zt^2)\epsilon^2 + x(8x^2 - 2\epsilon^2), \quad f_2^{\text{BR}} = \sqrt{2}x(3zt^2 - t^3) + y(8x^2 - \epsilon^2). \qquad (15)$$

The $S_\epsilon^3$-constraint is explicitly given by the restriction $\epsilon^2 = x^2 + y^2 + z^2 + t^2$ for some small constant $\epsilon$, making sure that we obtain functions of three variables. This ensures that the common zeros of $f_1^{\text{F8}}$ and $f_2^{\text{F8}}$ represent the figure-eight knot, and the zeros of $f_1^{\text{BR}}$ and $f_2^{\text{BR}}$ represent the Borromean rings. The lattice momentum then has to be introduced differently, depending on wether continuous or discrete lattice models are desired. This can be done in many ways, and what we present below is merely exemplifying how this can be accomplished.

Desired continuous model Hamiltonians can be constructed using the different ansätze for the **d**-vector discussed in Sec. 2.1 by introducing the lattice momentum as

$$x = \epsilon^2 - (k_x^2 + k_y^2 + k_z^2), \quad y = k_x, \quad z = k_y, \quad t = k_z. \qquad (16)$$

Conventional Hermitian Hamiltonians are obtained using Eq. (10) or Eq. (11), while Eq. (8) yields non-Hermitian models. The band structure of such a model hosting an exceptional figure-eight knot and Borromean rings are displayed in Fig. 1 (a) and (c) respectively, while their corresponding Fermi surfaces are shown in Fig. 1 (e) and (g).

These continuous models can be discretised if the lattice momentum is introduced periodically. Thinking about Eq. (16) as an expansion for small momentum, one naturally arrives at,

$$x = \epsilon^2 + \sum_{i=x,y,z} \cos(k_i) - 3, \quad y = \sin(k_x), \quad z = \sin(k_y), \quad t = \sin(k_z). \qquad (17)$$

The explicit forms of the functions whose common zeros represent a figure-eight knot read as

$$f_1^{\text{F8}} = \left(\epsilon^2 + \sum_{i=x,y,z} \cos(k_i) - 3\right)\left[8\left(\epsilon^2 + \sum_{i=x,y,z} \cos(k_i) - 3\right)^2 - 2\epsilon^2\right]$$
$$+ \epsilon^2\left(\sin^2(k_y) - \sin^2(k_z)\right), \qquad (18)$$

$$f_2^{\text{F8}} = \left[8\left(\epsilon^2 + \sum_{i=x,y,z} \cos(k_i) - 3\right)^2 - \epsilon^2\right]\sin(k_x)$$
$$+ 2\sqrt{2}\left(\epsilon^2 + \sum_{i=x,y,z} \cos(k_i) - 3\right)\sin(k_y)\sin(k_z). \qquad (19)$$

By constructing the **d**-vector corresponding to a non-Hermitian model Hamiltonian, as in Eq. (8), the highest appearing degree of the momentum components is three, meaning that the longest ranged hopping is three lattice constants. Its dual Hermitian model will have twice as large hopping distance, since the functions $f_1$ and $f_2$ are essentially squared in the model defined by Eq. (10). The PT-symmetric Hermitian model defined by Eq. (11) is instead linear in terms of $f_1$ and $f_2$, and thus its longest ranged hopping is the same as for the non-Hermitian model. This illustrates that non-Hermitian models are twice as local as their dual Hermitian models, while symmetries can be used to obtain Hermitian models with the same hoping distance. The band structure of a non-Hermitian tight-binding model hosting an exceptional figure-eight knot is displayed in Fig. 1 (b), while Fig. 1 (f) shows its Fermi surface.
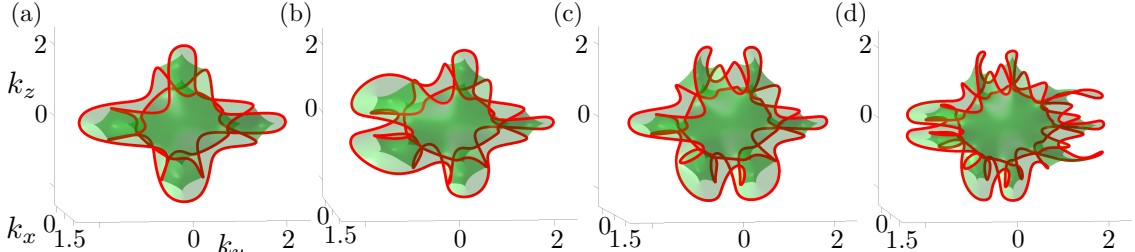

Figure 2: Illustration of more complex hyperbolic exceptional structures, obtained from lattice models defined by Eq. (8), Eq. (12) and Eq. (17) with $\epsilon = 1$, $\Lambda = 10^4$ and $n = 4, 5, 6, 9$ respectively. $\Lambda$ is taken to be large in order to smoothen the exceptional structures. These structures resembles the $(3, n)$ Turk's head links for their respective $n$, which means that the structures in (a) and (b) are knots, while those in (c) and (d) both are Brunnian links. All this is verified by calculating their respective Alexander polynomials, which are presented in Sec. 3.

Analogously, the functions whose common zeros represent the Borromean rings becomes

$$
f_1^{\text{BR}} = \left( \epsilon^2 + \sum_{i=x,y,z} \cos(k_i) - 3 \right) \left[ 8 \left( \epsilon^2 + \sum_{i=x,y,z} \cos(k_i) - 3 \right)^2 - 2\epsilon^2 \right]
$$
$$
+ \epsilon^2 \left( \sin^3(k_y) - \sin(k_y) \sin^2(k_z) \right), \tag{20}
$$

$$
f_2^{\text{BR}} = \left[ 8 \left( \epsilon^2 + \sum_{i=x,y,z} \cos(k_i) - 3 \right)^2 - \epsilon^2 \right] \sin(k_x)
$$
$$
+ 2\sqrt{2} \left( \epsilon^2 + \sum_{i=x,y,z} \cos(k_i) - 3 \right) \left( 3 \sin^2(k_y) \sin(k_z) - \sin^3(k_z) \right). \tag{21}
$$

In the same fashion as before, we can read off that the longest ranged hopping for the PT-symmetric Hermitian model defined by Eq. (11) and the non-Hermitian model defined by Eq. (8) will be four lattice constants, while the hopping for the dual Hermitian model described by Eq. (10) will be twice as large. Again, the band structure of such a non-Hermitian system is showed in Fig. 1 (d), with its corresponding Fermi surface in Fig. 1 (h).

It is worth stressing that only nodal links appearing in non-Hermitian structures bound the Fermi surface of the system, making this a feature unique for the non-Hermitian realm. This furthermore means that the Fermi surface coincides with a Seifert surface of the corresponding link [14, 18].

## 2.3 More General Turk's Head Links and Topological Transitions

The polynomial in Eq. (12) can be used to generate Hamiltonians with even more exotic nodal knots and links by considering larger values of $n$. In Fig. 2 we display some additional examples of exceptional nodal knots and links obtained by the method described above. In contrast to the figure-eight knot and the Borromean rings previously discussed, there is to our knowledge no explicit result telling us what knots or links to expect for larger $n$. By the methods introduced in Sec. 3 these structures are identified, and an initial pattern is observed, suggesting that for a general $n$, the corresponding nodal structure is the $(3, n)$ Turk's head link. The $(p, q)$ Turk's head link is obtained by considering a diagram of the $(p, q)$ torus link, and making it alternate, i.e., changing the crossings such that an under crossing is always followed by an over crossing

and vice versa [28]. As for torus links, the $(p,q)$ Turk's head link will be a knot if $p$ and $q$ are relatively prime, and if $\text{GCD}(p,q) = m$, it is a link of $m$ $\left(\frac{p}{m}, \frac{q}{m}\right)$ Turk's head knots. It is noteworthy that the $(3,n)$ Turk's head link is a Brunnian link of three components when $n$ is a multiple of 3.

Interestingly enough, this construction includes some transitions depending on the radius $\epsilon$ of the three sphere. Generally, these features are that for a given $n$ and a small radius $\epsilon = \delta$, the nodal structure will be a $(2,n)$ torus link. When increasing $\epsilon$ a circle centered at the origin will appear at some $\epsilon = r_1$. When increasing $\epsilon$ further, this circle will grow and eventually, at some $\epsilon = r_2$, it will join with the torus link, forming the $(3,n)$ Turk's head link. In Fig. 3 we illustrate these transitions for the simplest case $n = 2$. As a general comment of these transition, we note that the complexity of the nodal knot increases when undergoing these transitions for growing $\epsilon$. In particular, the genus of the nodal structure increases. This in turn means that the minimal genus of the corresponding Fermi surface also will increase, since it coincides with a Seifert surface of the knot or link. This can potentially be experimentally observed in spectroscopical systems, e.g. with light scattering experiments in phononic crystals [29].

## 3 Knot Invariants

In contrast to nodal points, such as Weyl- and Dirac points, and unknotted nodal lines, i.e. (possibly linked) circles, which are characterised by integer valued invariants such as Chern number and linking numbers, the invariants of knots attain the form of polynomials. So far, such polynomials have not been calculated for a general model Hamiltonian. In this section, we provide such an algorithm for computing the Alexander polynomial, by using an approximate, yet topologically equivalent, explicit parameterisation of the nodal structures. This polynomial characterises the knottedness of the nodal structures, providing a way to determine their topologies without visual inspection.

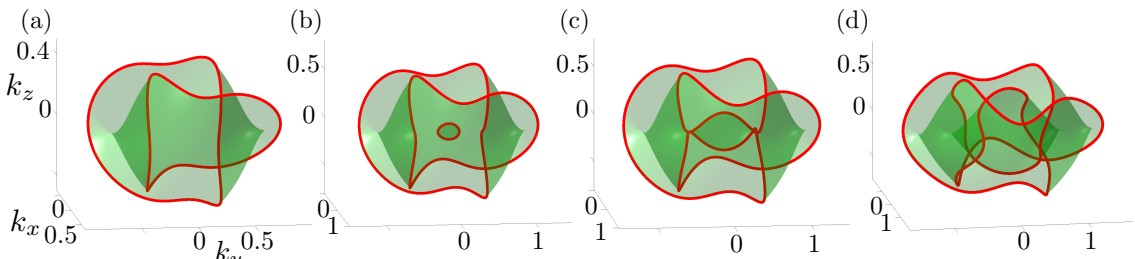

Figure 3: Illustration of how the transitions in a lattice model defined by Eq. (8), Eq. (12) and Eq. (17) take place when $n = 2$ with $\Lambda = 100$. In (a) $\epsilon = 0.1$ which results in a $(2,2)$ torus link, the Hopf link. Eventually, a new circle appears in the center, which is illustrated in (b) for $\epsilon = 0.26$. This circle merges with the Hopf link when $\epsilon$ is allowed to grow, and the critical value $\epsilon = 0.303$ is displayed in (c). For larger $\epsilon$, the figure-eight knot, or more generally, the $(3,n)$ Turk's head knot is obtained, displayed in (d) for $\epsilon = 1$.

### 3.1 Parameterisation

Since the nodal structures are obtained by intersecting two implicitly defined surfaces, $\text{Re}[E^2] = 0$ and $\text{Im}[E^2] = 0$ (or $f_1 = 0$ and $f_2 = 0$ in the PT-symmetric Hermitian case), it is generally hard to analytically determine an explicit function representing the intersection curve. Numerically it is however possible to find an ordered collection of points belonging to

the curve. From this, we compute an approximate, yet topologically equivalent curve using an ansatz in terms of Fourier basis functions. If we denote by $\mathbf{k}(\tilde{s})$ the discrete collection of points with components $k_i(\tilde{s})$, $i = x, y, z$, and by $\phi_j(\tilde{s})$ the Fourier basis functions

$$\phi_{2l}(\tilde{s}) = \sin(l\tilde{s}), \quad \phi_{2l+1}(\tilde{s}) = \cos(l\tilde{s}), \quad l \in \mathbb{N}, \tag{22}$$

where $\tilde{s}$ denotes a discrete parameter, attaining symmetrically spaced values in the closed interval $[0, 2\pi]$. The number of values $\tilde{s}$ can attain is equal to the number of discrete points we know on the nodal structure. Then, we make the ansatz,

$$k_i(\tilde{s}) = \sum_{j=1}^{2j_{\max}+1} C_{ij} \phi_j(\tilde{s}), \tag{23}$$

which we can write in matrix form as

$$\mathbf{k}(\tilde{s}) = \mathbf{C} \cdot \boldsymbol{\phi}(\tilde{s}). \tag{24}$$

By solving the system of equations given by Eq. (24), we get a matrix of coefficients $\mathbf{C}$. If we now make an analogous, but continuous, ansatz,

$$k_i(s) = \sum_{j=1}^{2j_{\max}+1} C_{ij} \phi_j(s), \tag{25}$$

where $s \in [0, 2\pi]$ is a continuous parameter and $C_{ij}$ are the coefficients obtained by solving Eq. (24), we get a continuous curve described by $k_i(s)$, whose topology coincides with that of the nodal structure for sufficiently large $j_{\max}$. The resulting curve can be put on a more convenient form by replacing $\phi_j(s)$ by its definition, giving,

$$k_i(s) = \gamma_i + \sum_{j=1}^{j_{\max}} \alpha_{ij} \cos(js) + \beta_{ij} \sin(js), \tag{26}$$

where $\gamma_i$ are the elements in the first column of $\mathbf{C}$, $\alpha_{ij}$ are the elements of the odd columns of $\mathbf{C}$, except for the first, and $\beta_{ij}$ are the elements of the even columns of $\mathbf{C}$. In Sec. 3.5, we argue that this expansion remains stable for large $j_{\max}$.

For relatively simple objects, this series can be terminated rather quickly and still provide sufficient information. The Borromean rings only require $j_{\max} = 2$ while the trefoil and figure-eight knots require $j_{\max} = 3$ and $j_{\max} = 4$ respectively in order to be approximated well enough that the topology is faithfully captured (cf. Figure 4 and Table 3). As an explicit quantitative example we give numerical details on how this works out in the case of the nodal trefoil knot obtained from the continuum model in Ref. [18]: using e.g. the Matlab-function *isocurve3.m* [30], defining the coordinates $k_x$, $k_y$ and $k_z$ as an evenly distributed $300 \times 300 \times 300$-grid between $-0.8$ and $0.8$, results in 3943 ordered points on the curve. The corresponding parameterisation coefficients are explicitly given by,

$$\boldsymbol{\alpha} = \begin{pmatrix} -0.0430 & -0.4262 & -0.0499 \\ -0.0709 & -0.2488 & 0.0742 \\ 0.0144 & -0.0142 & -0.2133 \end{pmatrix},$$

$$\boldsymbol{\beta} = \begin{pmatrix} -0.2475 & 0.2276 & -0.0165 \\ -0.0440 & -0.4466 & -0.0842 \\ -0.0072 & -0.0555 & 0.1942 \end{pmatrix}, \quad \boldsymbol{\gamma} = \begin{pmatrix} 0.0222 \\ 0.0079 \\ -0.0028 \end{pmatrix}, \tag{27}$$

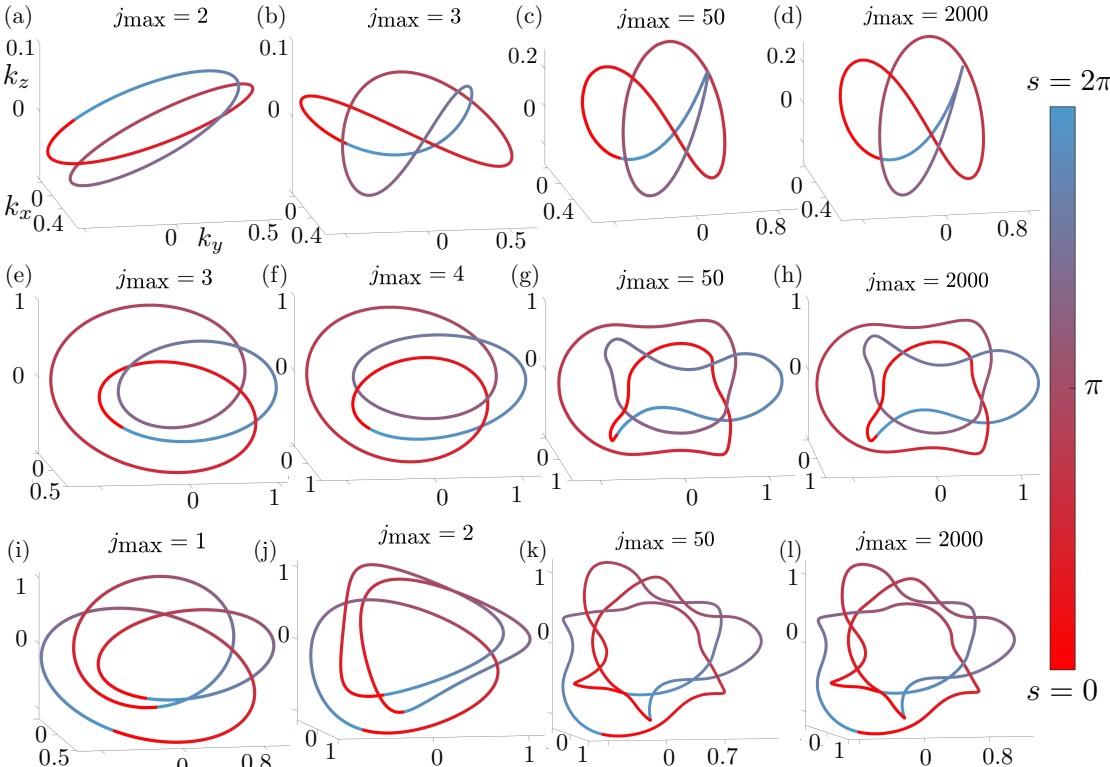

Figure 4: An illustration of the transitions occurring for the parameterised curves. For small $j_{\mathrm{max}}$ the topology of the parameterisation does not resemble that of the nodal structure, but when considering additional modes it will eventually transition towards the correct topology. In (a)-(d), parameterisations of a trefoil for different $j_{\mathrm{max}}$ are shown, with a transition taking place at $j_{\mathrm{max}} = 3$ – the curve in (b) is a trefoil knot, while (a) shows an unknot. In (e)-(h) the figure-eight knot is approximated. Interestingly enough, the curve in (e) actually resembles a trefoil knot, while (f)-(h) inherit the topology of a figure-eight knot. The lower bar (i)-(l) represents parameterisations of the Borromean rings, where the transition to the correct topology occurs already for $j_{\mathrm{max}} = 2$. Noteworthy is that the structure in (i) consist of three individually linked circles, which are unlinked for larger $j_{\mathrm{max}}$. The topological transitions occurring for these structures are observed by computing the corresponding Alexander polynomial, which are presented in Tab. 3.

where $\boldsymbol{\alpha}, \boldsymbol{\beta}$ and $\boldsymbol{\gamma}$ are matrices with elements $\alpha_{ij}, \beta_{ij}$ and $\gamma_i$ respectively. Examples of parameterised curves for different $j_{\mathrm{max}}$ and targeted knots and links are shown in Fig. 4. The number of required modes increases with the complexity of the nodal structures. This explicit embedding enables calculations of knot and link invariants as we turn to next.

We note that a similar approximation ansatz is used in order to parameterise knots in Ref. [31]. However, their goal is to use this to calculate explicit functions whose zeros represent arbitrary knots and links, while we use it to approximate the set of zeros of such a function in order to calculate knot invariants for the band structure of a given model Hamiltonian.

## 3.2 Alexander Polynomial

In knot theory, the most commonly appearing knot invariants are the different knot polynomials. One of many such is the Alexander polynomial $\Delta_K(t)$ [32]. Even though some different

knots are known to share the same Alexander polynomial, it is of great help when knots are to be distinguished [33,34]. There are several ways of calculating $\Delta_K(t)$, and below we will use two of them. Both methods uses the determinant of the Alexander matrix, making them rather efficient from a computational point of view. Conventionally, knot polynomials, such as the Jones polynomial and HOMFLY-polynomial, are calculated using local surgeries on the crossings – so-called *skein relations*. This procedure quickly becomes complicated, as a diagram with $n$ crossings results in $2^n$ diagrams to take into consideration – the complexity of the calculation grows exponentially with the complexity of the knot. For an equivalent problem, the Alexander polynomial can be obtained by computing the determinant of an $(n-1) \times (n-1)$-matrix. By implementing LU-decomposition, this is performed by doing only $\mathcal{O}(n^3)$ computations, making it a much simpler problem to solve. Since none of the polynomials mentioned above can distinguish every knot [35], the efficiency of the calculations makes the Alexander polynomial attractive for practical calculations.

It should be noted that for relatively simple knots there are tables listing essentially all the information there is to know about them, including the Alexander polynomial. However, for more exotic knots the list of similar tables grows thin, but in Ref. [28] an algorithm calculating the Alexander polynomial for a general $(p,q)$ Turk's head knot is presented, making it possible to identify our new nodal structures for any $n$.

### 3.2.1 Alexander Polynomial from the Knot Parameterisation

The method for constructing the Alexander matrix **A** from the knot itself goes as follows, further details are found in [36].

1. Choose an orientation of the knot.

2. Label all crossings $c_1, ..., c_n$.

3. Label all arcs $a_1, ..., a_n$, starting from either an under- or an over crossings.

4. Determine the signature of the crossings, using the conventions illustrated in Fig. 5.

5. Assign matrix elements.

    (a) If crossing $l$ is positive , assign

    $$A_{li} = 1 - t, \quad A_{lj} = -1, \quad A_{lk} = t. \tag{28}$$

    (b) If crossing $l$ is negative, assign

    $$A_{li} = 1 - t, \quad A_{lj} = t, \quad A_{lk} = -1. \tag{29}$$

If any of $i, j, k$ are equal, both contributions are added and assigned to the corresponding position in the matrix. The result will be an $(n \times n)$-matrix. Remove the last row and column, and take its determinant. The result is a polynomial $\tilde{\Delta}_K(t)$, which is related to the Alexander polynomial by $\pm t^k \tilde{\Delta}_K(t) = \Delta_K(t)$, with $k \in \mathbb{Z}$. The exponent $k$ and the overall sign are determined by the constraints

$$\Delta_K(t) = \Delta_K(t^{-1}), \quad \Delta_K(1) = 1. \tag{30}$$

By scaling $\tilde{\Delta}_K(t)$ such that this is fulfilled, $\Delta_K(t)$ is the Alexander polynomial.

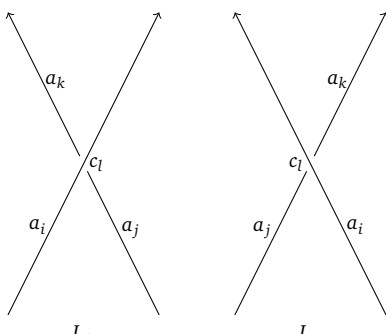

Figure 5: Illustration of the convention of the signatures of the crossings, where the arrows indicates the orientation. $L_+$ and $L_-$ are what we refer to as positive and negative crossings respectively. Generally, $k = j + 1$ due to how the arcs are defined. This is of course not true when $k = 1$.

Table 1: Alexander polynomials for a few different nodal structures. The left column is what comes out from the algorithm, while the middle column is the factor which is to multiply the determinant in order to obtain the correctly scaled Alexander polynomial. Using these polynomials, we can determine what kind of nodal knot we have, without using graphical tools.

| Determinant $\tilde{\Delta}_K(t)$ | Scale Factor | Corresponding Knot |
|:---:|:---:|:---:|
| $-t^2 + t - 1$ | $-t^{-1}$ | Trefoil, $T_{(3,2)}$ |
| $-t^4 + t^3 - t^2 + t - 1$ | $-t^{-2}$ | Cinquefoil, $T_{(5,2)}$ |
| $t^9 - t^8 + t^6 - t^5 + t^4 - t^2 + t$ | $t^{-5}$ | $T_{(5,3)}$ |
| $t^6 - t^5 + t^4 - t^3 + t^2 - t + 1$ | $t^{-3}$ | Septfoil, $T_{(7.2)}$ |
| $t^8 - t^7 + t^6 - t^5 + t^4 - t^3 + t^2 - t + 1$ | $t^{-4}$ | $T_{(9,2)}$ |
| $t^5 - 3t^4 + t^3$ | $-t^{-4}$ | Figure-Eight Knot |
| $-t^{19} + 5t^{18} - 10t^{17} + 13t^{16} - 10t^{15} + 5t^{14} - t^{13}$ | $t^{-16}$ | $TH_{(3,4)}$ |
| $t^{20} - 6t^{19} + 15t^{18} - 24t^{17} + 29t^{16}$ $-24t^{15} + 15t^{14} - 6t^{13} + t^{12}$ | $t^{-16}$ | $TH_{(3,5)}$ |

In order to implement this algorithm numerically, we proceed as follows. First of all, we need a 2D projection $P$ of the parameterised knot. Here, we choose to project onto the plane $\pi : k_z = -1$ along its unit normal. Thus, the projection $P$ reads

$$P : \begin{pmatrix} k_x(s) \\ k_y(s) \\ k_z(s) \end{pmatrix} \twoheadrightarrow \begin{pmatrix} k_x(s) \\ k_y(s) \end{pmatrix}, \tag{31}$$

where $k_i(s)$, $i = x, y, z$ are given by the Fourier ansatz Eq. (26). The orientation of the knot is naturally obtained by increasing the parameter $s \in [0, 2\pi]$. Note though that the orientation is not in any way naturally induced by the band intersection. It is a pure artifact of the chosen parameterisation of the curve, and could equally well have been reversed. Then we calculate the self intersections of the projection. Note that if we get anything else than ordinary double points, i.e., if we have points where more than two arcs intersect, or if the arcs only touch each other, we need to choose a different projection. If we have good intersections, their signature is determined by considering the slope and orientation of the intersecting arcs, see Fig. 5. Lastly, the arcs are labeled, where we use the convention that one arc starts at an under crossing, and ends at the next, in the direction of the orientation. For every crossing, we check which are the neighbouring arcs, and assign elements to the matrix according to the rules. Some examples

Table 2: Alexander polynomials for a few different exceptional structures. The left column is what comes out from inspection of the Fermi surface, using the algorithm mentioned in Section 3.2.2, while the middle column is the is the factor which is to multiply the determinant in order to obtain the correctly scaled Alexander polynomial. Only by considering the Fermi surface, the topology of the knotted structures can be determined.

| Determinant | Scale Factor | Corresponding Knot |
|---|---|---|
| $t^2 - t + 1$ | $t^{-1}$ | Trefoil, $T_{(3,2)}$ |
| $t^4 - t^3 + t^2 - t + 1$ | $t^{-2}$ | Cinquefoil, $T_{(5,2)}$ |
| $t^8 - t^7 + t^5 - t^4 + t^3 - t + 1$ | $t^{-4}$ | $T_{(5,3)}$ |
| $t^6 - t^5 + t^4 - t^3 + t^2 - 1 + 1$ | $t^{-3}$ | Septfoil, $T_{(7,2)}$ |
| $t^8 - t^7 + t^6 - t^5 + t^4 - t^3 + t^2 - t + 1$ | $t^{-4}$ | $T_{(9,2)}$ |
| $-t^2 + 3t - 1$ | $t^{-1}$ | Figure-Eight-Knot |

of polynomials we obtain numerically, together with the appropriate scale factors, are given in Table 1.

### 3.2.2 Alexander Polynomial from the Fermi Surface

An alternative approach for obtaining the Alexander polynomial is to visually inspect a Seifert surface of the knot. This is physically meaningful since the Fermi surface of a non-Hermitian system is a Seifert surface of the corresponding exceptional structure. From a Seifert surface, a Seifert matrix $S$ can be constructed, which is related to the corresponding Alexander matrix by $A = S - tS^T$. We will not describe the exact method in detail, and refer to [37] for details.

The key point is to deform the Fermi surface to discs connected with half-integer twist ribbons. This is diagrammatically represented by discs, connected with labelled lines, where the labels denote the number and orientation of half-twists of the ribbons. This in turns needs to be simplified until we reach a topologically equivalent diagram with only one disc and a bunch of lines which are labeled $b_1, ..., b_n$. The number of full twists related to every $b_i$ gives us the diagonal elements $S_{ii}$ in the matrix. How the lines connecting the twist-labels cross each other determines the off-diagonal elements. If band $b_l$ crosses over band $b_j$ $m$ times from left to right, $S_{lj} = m$, and if it crosses over $m$ times from right to left, $S_{lj} = -m$. If there is no crossing, $S_{lj} = 0$. By taking the determinant of $S - tS^T$, we get a polynomial, which scaled properly is the Alexander polynomial. Some calculations are shown in Table 2.

The downside of this method is that it relies on visual inspection instead of direct calculation, making it rather involved as the knot gets more complicated. Also, it is only applicable for non-Hermitian systems. Still, the fact that we are able to determine the topology of the exceptional structure by only using something as physical as the Fermi surface, makes it worth discussing.

### 3.3 Linking- and Self Linking Number

The approximate, yet topologically equivalent, parameterisation of the nodal structure provides a direct way of calculating linking numbers, making it possible to check if and how the structures are linked together. The Gauss' linking formula reads [38],

$$\text{lk}_{\mathcal{C}_1, \mathcal{C}_2} = \frac{1}{4\pi} \int_{\mathcal{C}_1} \int_{\mathcal{C}_2} ds\, ds'\, \epsilon_{\mu\nu\alpha} \frac{dx^\mu(s)}{ds} \frac{dy^\nu(s')}{ds'} \frac{x^\alpha(s) - y^\alpha(s')}{|\mathbf{x}(s) - \mathbf{y}(s')|^3}, \tag{32}$$

where $s$ and $s'$ are parameters, $\mathbf{x}(s)$ and $\mathbf{y}(s')$ are coordinates on curve $\mathcal{C}_1$ and $\mathcal{C}_2$ respectively with components $x^\mu(s)$ and $y^\mu(s')$, and $\epsilon_{\mu\nu\alpha}$ is the completely anti-symmetric Levi-Civita sym-

bol. Note that repeated indices are summed over. With the approximative curves, we get, e.g., the following

$$|\text{lk}_{2,2}| = 1, \quad |\text{lk}_{4,2}| = 2, \quad |\text{lk}_{6,4}| = 6, \quad |\text{lk}_{\text{BR}}| = 0, \tag{33}$$

where the integral is numerically evaluated using $\mathcal{O}(10^3)$ number of points on the curves, yielding an estimated error of approximately $10^{-4}$. We only present absolute values of the linking number, since its sign is affected by the orientation of the respective components. Recall that the orientation of the parameterised nodal structure is not given by the system itself, making it a technical artifact with no physical meaning. $\text{lk}_{\text{BR}} = 0$ means that all pairwise linking numbers for the structures displayed in Fig. 1 (b) and (d) vanish. $\text{lk}_{p,q}$ denotes the linking number for the $(p, q)$-torus link. By also calculating the Alexander polynomial for the individual components of the link, we can determine the topology of the link without graphically considering it.

The self linking number for knots is however a far more complicated story. Recall that it is to be thought of as the linking number between a knot $K$ and its push-off $K'$, where $K'$ is defined by displacing $K$ a small distance along a nowhere tangent vector field, a so-called *framing*. For a given choice of framing, the self linking number is invariant, while changing framing can yield any result [5]. A specific choice of framing that would be physically meaningful is the gradient of the surface $\text{Re}[E^2] = 0$ or $\text{Im}[E^2] = 0$ evaluated along the knot. However, it is clear that when cutting along the knot, both $\text{Re}[E^2] = 0$ and $\text{Im}[E^2] = 0$ are split into two Seifert surfaces respectively. Thus, the framing defined by the gradient evaluated along the knot, call it $u$, consist of vectors normal to both the surface and the knot. This is homotopic to a framing consisting of vectors tangent to the surfaces, but normal to the knot – a so-called *Seifert framing*. If we denote such a framing by $v$, the homotopy is explicitly given by

$$s \mapsto u\cos(s) + v\sin(s), \tag{34}$$

where $s = 0$ gives $u$, and $s = \frac{\pi}{2}$ gives $v$. Homotopic framings yield the same self linking number for a given knot, and it is well-known, but not explicitly stated in the literature, that the self linking number for a Seifert framing is always zero. Here is a sketch of the argument. Suppose $K \subset \mathbb{R}^3$ is a knot bounding an oriented surface $S$. Let us suppose that for all points $x \in K$, $v(x)$ is the unit vector that is tangent to $S$ and orthogonal to $K$ at $x$, and is pointing toward $S$. Thus $v$ is a Seifert framing of $K$. Let $K'$ be the knot obtained by displacing $K$ a short distance along $v$. The inclusion $K' \subseteq \mathbb{R}^3 \setminus K$ induces a homomorphism of first homology groups. Both homology groups are isomorphic to $\mathbb{Z}$, so the homomorphism is given by a multiplication by an integer (determined up to sign). This integer is the the self linking number of $K$ with the framing $v$. The point is that the inclusion $K' \hookrightarrow \mathbb{R}^3 \setminus K$ factors through the inclusion $K' \hookrightarrow S'$, where $S'$ is a surface obtained from $S$ by removing a collar neighborhood of its boundary $K$, and $K'$ is the boundary of $S'$. The inclusion of a connected $n - 1$-dimensional manifold as the boundary of an orientable $n$-dimensional manifold is always zero on homology in degree $n - 1$. We only use this fact for $n = 2$, and in this case it can be proved using the classification of surfaces. But the general case is proven in e.g. Chapter 21 of [39], to which we refer for further details. Thus, the framing defined by evaluating the gradient of $\text{Re}[E^2] = 0$ or $\text{Im}[E^2] = 0$ will not provide any information, since they are homotopic to a Seifert framing.

Another alternative would be to use the Frenet frame, consisting of the unit tangent vector of the knot, the unit normal vector and their cross-product. The disadvantage of this choice is that there might be points where the tangent vector vanishes, so-called inflection points. For such points, the Frenet frame, and thus the corresponding self linking number, are not defined at all, and thus the Frenet frame will not provide sufficient information. Thus, we conclude that the self linking number does not qualify as a useful knot invariant to be calculated.

### 3.4 Higher Order Milnor Invariants

Even though the pariwise linking numbers between all the components in the nodal structures displayed in Fig. 1 (b) and (d) vanish, we have been claiming that they are nontrivial links. This can be proven by calculating higher order Milnor invariants. These are widely used in knot theory in order to characterise the family of Brunnian links – links that are nontrivial even though all components are pairwise unlinked. There is a nice geometrical interpretation of these invariants, making them obtainable from the constants appearing in the Conway polynomial $\nabla_L(z)$ of the link [40]. This knot invariant polynomial is intimately related to the Alexander polynomial $\Delta_L(z)$ through the relation

$$\nabla_L(z - z^{-1}) = \Delta_L(z^2). \tag{35}$$

Thus, Milnor invariants can be computed by using the algorithm calculating the Alexander polynomial described in Sec. 3.2.1, and then solving Eq. (35) for the Conway polynomial. The explicit relation between the constants and the Milnor invariants varies depending on the number of components in the link, and we refer to [40] for further details.

To illustrate this, we apply this technique to calculate the Milnor invariant $\mu(123)$ – the triple linking number – for the nodal structures in Fig. 1 (b) and (d). The algorithm yields the following polynomials,

$$\tilde{\Delta}_{\mathrm{BR}}^{\mathrm{LM}}(t) = t^7 (t-1)^4, \tag{36}$$

$$\tilde{\Delta}_{\mathrm{BR}}^{\mathrm{Cont}}(t) = t^2 (t-1)^4. \tag{37}$$

Note that in order to obtain something trustworthy for the continuous model, we have changed the projection, since the one given by Eq. (31) almost resulted in a triple intersection point. The projection used for the continuous model, yielding Eq. (37), is instead given by,

$$\tilde{P} : \begin{pmatrix} x(s) \\ y(s) \\ z(s) \end{pmatrix} \twoheadrightarrow \begin{pmatrix} x(s) \\ z(s) \end{pmatrix}. \tag{38}$$

Both Eq. (36) and Eq. (37) become, when scaled correctly,

$$\Delta_{\mathrm{BR}}(t) = t^2 - 4t + 6 - 4t^{-1} + t^{-2}. \tag{39}$$

Using Eq. (35), we see that,

$$\nabla_{\mathrm{BR}}(z - z^{-1}) = z^4 - 4z^2 + 6 - 4z^{-2} + z^{-4}, \tag{40}$$

which results in,

$$\nabla_{\mathrm{BR}}(z) = z^4. \tag{41}$$

According to [40], a link with three components has a Conway polynomial of the form,

$$\nabla_L(z) = z^2 (a_0 + a_2 z^2 + ...), \tag{42}$$

where $a_{2i}$ are real constants. That $a_0$ in our case vanishes is related to that all our pairwise linking number vanishes [40]. Furthermore, given that $a_0 = 0$, then $a_2$ is related to the triple linking number as $a_2 = (\mu(123))^2$. Thus, Eq. (41) says that the triple linking numbers for the structures in Fig. 1 (b) and (d) are $\pm 1$, proving that the links are nontrivial and more specifically, that they are equivalent to the Borromean rings.

In Sec 2.3 we furthermore claimed that Fig. 2 (c) and (d), corresponding to $n = 6$ and $n = 9$ respectively, also are Brunnian links, and we show that in the same way as for the

Borromean rings. The algorithm yields the following polynomials for the structures in Fig. 2 (c) and (d).

$$\tilde{\Delta}_{n=6}(t) = t^{15}(t-1)^4(1 - 4t + 10t^2 - 4t^3 + t^4) \tag{43}$$

$$\tilde{\Delta}_{n=9}(t) = t^{14}(t-1)^4(1 - 4t + 9t^2 - 4t^3 + t^4)^2, \tag{44}$$

which by a straight-forward calculation yield the following Conway polynomials,

$$\nabla_{n=6}(z) = z^2(4z^2 + z^6) \tag{45}$$

$$\nabla_{n=9}(z) = z^2(9z^2 + 6z^6 + z^{10}), \tag{46}$$

from where we see that all the individual linking numbers vanish, and that the triple linking number is ±2 and ±3 for $n = 6$ and $n = 9$ respectively.

This useful geometrical interpretation of the Conway polynomial, or equivalently, the Alexander polynomial, is an additional reason to compute the Alexander polynomial instead of the Jones polynomial. This provides a way to not only see whether or not links are nontrivial, but also what class of links they belong to. Concretely, this method classifies all nodal links up to homotopy, since that is exactly what the Milnor invariants do. We note that the same information can be extracted from the more general HOMFLY-polynomial, but as mentioned in Sec. 3.2, the complexity of the algorithms that have to be implemented makes the Alexander polynomial favourable.

## 3.5 Topological Transition and Convergence of the Parameterisation

The parameterisation ansatz Eq. (26) only describes curves topologically equivalent to the nodal knots when $j_{\max}$ is sufficiently large. In Tab. 3 we show this using two examples – the trefoil and figure-eight knot respectively. If $j_{\max}$ is too small, the correct topology is not achieved, but when increasing $j_{\max}$, it eventually transition to the correct one. A question that needs to be adressed though is if this type of expansion remains stable when $j_{\max}$ grows. Again, we recall that the knots (links) are obtained as the intersection between two surfaces. In the case of any knotted (linked) structure dealt with here, these surfaces are described as zeros of smooth functions in three variables, resulting in two smooth 2D submanifolds of $\mathbb{R}^3$ or $\mathbb{T}^3$. These submanifolds intersect each other transversally, meaning that their normal vectors are not collinear at the intersection. Thus, the intersection is itself a (collection of) smooth manifold(s) of dimension(s) $2 + 2 - 3 = 1$, i.e. a (collection of) smooth and closed curve(s) in $\mathbb{R}^3$ or $\mathbb{T}^3$. Each component can be represented by a smooth embedding $g : S^1 \to \mathbb{R}^3$ or $\mathbb{T}^3$ with $s \mapsto \left( g_{k_x}(s), g_{k_y}(s), g_{k_z}(s) \right)$, with corresponding Fourier series

$$g_i(s) = c_i + \sum_{j=1}^{\infty} \left( a_{ij} \cos(js) + b_{ij} \sin(js) \right), \tag{47}$$

where the convergence of the Fourier series is ensured since $g$ is a smooth periodic function. Moreover, the Fourier series can be differentiated term by term, and the series of derivatives converges as well. This means that the Fourier series converges in Whitney $C^1$ topology. Comparing this to the approximate parameterisation Eq. (26) in the limit $j_{\max} \to \infty$, we see that they coincide. Thus, the approximate parameterisation of the nodal knots given by Eq. (26) converges to the Fourier series Eq. (47) when $j_{\max} \to \infty$, which in turn converges to the embedding describing the intersection of the two smooth submanifolds. Therefore, the approximation remains stable when additional modes are considered, ensuring preservation of its topology when $j_{\max} \to \infty$.

Table 3: A concrete illustration of how the topology of the parameterised curve changes with the number of modes considered. When considering large enough $j_{max}$, the ansatz Eq. (26) provides a curve topologically equivalent to the nodal structures, but when considering too few modes, this is not acquired. At some points, topological transitions occur, eventually resulting in the desired topology. When the number of modes is taken to infinity, the ansatz converges to the Fourier series of the smooth embedding describing the intersection curve, which in turn converges to the embedding itself, ensuring stability of the ansatz in Eq. (26). Note that we here use a different projection, namely onto the plane $x = -1$ along its unit normal, when calculating the polynomials for the Borromean rings.

| $j_{max}$ | Trefoil Polynomial | Figure-Eight Polynomial | Borromean Rings Polynomial |
|---|---|---|---|
| 1 | 1 | 1 | $t - 2 + t^{-1}$ |
| 2 | 1 | 1 | $t^2 - 4t + 6 - 4t^{-1} + t^{-2}$ |
| 3 | $t - 1 + t^{-1}$ | $t - 1 + t^{-1}$ | $t^2 - 4t + 6 - 4t^{-1} + t^{-2}$ |
| 4 | $t - 1 + t^{-1}$ | $-t + 3 - t^{-1}$ | $t^2 - 4t + 6 - 4t^{-1} + t^{-2}$ |
| 50 | $t - 1 + t^{-1}$ | $-t + 3 - t^{-1}$ | $t^2 - 4t + 6 - 4t^{-1} + t^{-2}$ |
| $\infty$ | $t - 1 + t^{-1}$ | $-t + 3 - t^{-1}$ | $t^2 - 4t + 6 - 4t^{-1} + t^{-2}$ |

# 4 Experimental Relevance

Despite their apparent complexity, the nodal knotted band structures discussed in this work may potentially be realised in a variety of platforms ranging from optical setups [41–43], mechanical systems [44] and cold atoms [45] to complex materials [46–51] in which nodal line degeneracies of various types such as rings, chains and links have been predicted or even realised. Starting from such ring or chain configurations it is conceivable that they can be tuned into more complex knotted or linked degeneracies by moderate deformations of the system, e.g. as very recently discussed in Ref. [52]. Deformations of this kind are particularly relevant to consider in the non-Hermitian realm where nodal line degeneracies are generic, i.e. they do not rely on any symmetries and any small deformation would leave them intact [14]. While non-Hermitian nodal chains, links and knots are still theoretical concepts, it is encouraging that exceptional rings have been reported in optical waveguide systems [42].

We also note that the simplest non-Hermitian links require only nearest neighbour hopping making them very realistic in dissipative photonic lattices [18]. While the longer range hopping needed for the new hyperbolic knots and links reported here make them more challenging to realise, very encouraging progress in engineering longer range hopping terms in a controlled fashion in photonic lattices has been reported [53].

The experimental signatures of nodal line semimetals, e.g. their 2D drumhead surface states, has recently lead to their experimental discovery in ARPES measurements on $ZrB_2$ [49] and in phononic crystals [54]. It is natural to think that similar surface states would provide useful information in the case of nodal knots as well, and this seems attractive given powerful detection methods such as ARPES and spin-resolved transport [55]. These surface states will however not provide full information of the knottedness of the systems, since that is a feature unique to 3D—the surface states could equally well origin from twisted unknots. In order to capture this one would in principle have to scan through all possible 2D projections, making it highly challenging. Instead, measuring the bulk band structure directly would be highly desirable and it is in this context worth emphasising that the Fermi-Seifert surfaces in the non-Hermitian realm should be detectable e.g. in photonic crystals where light scattering experiments have already revealed the existence on Fermi arcs induced by non-Hermiticity originating from losses [29].

Finally, we note that the while the 2D analogues with exceptional points and their concomitant Fermi arcs have been realised and observed photonic systems [29], they are predicted to also occur quite generally at material junctions [56], thus providing promising ingredients for realising the pertinent 3D nodal band structures in layered setups [57].

# 5  Conclusion

In this work, we have introduced Hermitian and non-Hermitian band structures hosting hyperbolic nodal structures including the figure-eight knot and the Borromean rings. While the Hermitian models rely on discrete symmetries, such as PT symmetry, the exceptional structures in the non-Hermitian realm are generic, making them stable towards any small perturbation. Moreover, the non-Hermitian nodal knots are intimately connected to novel open nodal Fermi surfaces, in the form of Seifert surfaces terminating on the exceptional nodal knots. To determine the topology of any nodal structure, we have developed an efficient and well-controlled method to calculate the corresponding Alexander polynomial from an approximate, but topologically equivalent and stable, Fourier basis curve. From this, we have shown that Milnor invariants of any order can be obtained to classify any linked structure up to homotopy.

Even though these hyperbolic structures appear complicated, we have constructed relatively simple tight-binding models where they can be realised, opening up for their experimental discovery.

*Notes added:* While finalising this manuscript, a preprint containing a model for the figure-eight knot and a possible realisation thereof occurred [58]. Beyond the results of this back to back study, in our present work we additionally consider higher order links as well as constructive algorithms to calculate topological knot invariants characterising microscopic models.

Shortly after uploading the first version of this manuscript, two pre-prints focusing on experimental perspectives of knotted and line nodal structures respectively appeared on the arXiv [59, 60].

# Acknowledgements

We would like to thank Alexander Berglund and Thors Hans Hansson for useful discussions and Johan Carlström for related collaborations. M.S., G.A. and E.J.B. are supported by the Swedish Research Council (VR) and M.S., and E.J.B. are additionally supported by the Wallenberg Academy Fellows program of the Knut and Alice Wallenberg Foundation. J.C.B. acknowledges financial support from the German Research Foundation (DFG) through the Collaborative Research Centre SFB 1143 (project-id 247310070).

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
