# Peer review of "Hyperbolic Nodal Band Structures and Knot Invariants"

_SciPost Physics, doi:SciPost Phys. 7, 019 (2019)_

## Round 1 · Referee Report · Anonymous (Referee 1) · 2019-6-15

Strengths

  1. This manuscript is clear and well-written, and is of pedagogical value particularly in Sect. 3, where mathematical results on the Milnor invariant are explained to a physics audience that is probably new to such concepts.

  2. This work is self-contained in that the whole story, from model construction and approach to knot characterization, are all explained in sequential order.

Weaknesses

  1. This work is of limit novelty, since most of the results are not new at all. In Section 2, the microscopic models, or at least close variants of them, have already appeared in previous works ie. Refs 11 and 15 by most of the same authors. Section 3, while pedagogical, are mostly not new, being established mathematical results from knot theory.

  2. Even the approximation approach on truncating j_max, which did not appear in any previous works by the same authors, is not new. For instance, an analogous truncation approach for realizing arbitrary knots appeared in

Bode, Benjamin, and Mark R. Dennis. "Constructing a polynomial whose nodal set is any prescribed knot or link." arXiv preprint arXiv:1612.06328 (2016).

Report

As explained in the strengths and weakness remarks, this manuscript on the construction and characterizations of nodal knots is of limited novelty, even though it is well-written and of pedagogical value.

If I were to pick one aspect that is most interesting and original, it will be the determination of the Milnor invariant from the Conway polynomial.

Requested changes

  1. The authors should consider also citing some other existing/contemporary theoretical and experimental works on physical knots:

Bode, Benjamin, and Mark R. Dennis. "Constructing a polynomial whose nodal set is any prescribed knot or link." arXiv preprint arXiv:1612.06328 (2016). - please also contrast the approach used in your manuscript with this work.

Sugic, Danica, and Mark R. Dennis. "Singular knot bundle in light." JOSA A 35, no. 12 (2018): 1987-1999.

Li, Linhu, Ching Hua Lee, and Jiangbin Gong. "Boundary states of 4D topological matter: Emergence and full 3D-imaging of nodal Seifert surfaces." arXiv preprint arXiv:1905.07069 (2019).

Larocque, Hugo, Danica Sugic, Dominic Mortimer, Alexander J. Taylor, Robert Fickler, Robert W. Boyd, Mark R. Dennis, and Ebrahim Karimi. "Reconstructing the topology of optical polarization knots." Nature Physics 14, no. 11 (2018): 1079.

Zhang, Yi. "Cyclotron orbit knot and tunable-field quantum Hall effect." arXiv preprint arXiv:1905.02192 (2019). - which gives knotted analogies to cyclotron orbits

  1. In the note added pertaining to Ref 50, Ref 50 pertains to Hermitian and not non-Hermitian knots.

  2. It will be desirable to provide a more extensive discussion on how the approach used here deviates/complements the approaches of Refs 6 and 22, beyond the level of the relationship between d_R and d_I.

  • validity: high
  • significance: ok
  • originality: low
  • clarity: high
  • formatting: excellent
  • grammar: good

Author:  Marcus Stålhammar  on 2019-07-05  [id 555]

(in reply to Report 1 on 2019-06-15)

Dear Referee,

Thank you for reading the manuscript and for providing constructive suggestions for improvements and useful questions. Below, we have provided answers to all the points and questions raised in the report. Crucially we realise that our initial manuscript did lack clarity in what is original and how our work contrasts to earlier works. We hope that emphasising this in the updated manuscript version will clarify what is novel about our manuscript, thereby refuting the assessment that our findings are of low novelty.

Below we divide our answer in a fashion similar to the report, and answer the points raised under ”Weaknesses” and ”Requested Changes” accordingly.

Weaknesses

** 1. This work is of limit novelty, since most of the results are not new at all. In Section 2, the microscopic models, or at least close variants of them, have already appeared in previous works ie. Refs 11 and 15 by most of the same authors. Section 3, while pedagogical, are mostly not new, being established mathematical results from knot theory.**

It is true that related results have been obtained in Refs. 11 and 15 in the sense that the construction of the Hamiltonians is a natural extension of the methods of Ref. 8 and 15, which is indeed stated in the manuscript. However, it is not true that these microscopic models have appeared in the works referred to in the report. The new nodal structures presented here are in fact of a fundamentally different kind from the point of view of knots. In Ref 11, Hopf-links are presented, while in Ref 15 treats the general family of torus knots and links. These knots and links can be considered to be of the simplest sort, since they are completely classified from a mathematical point of view, see e.g. Milnor (1968). The figure-eight knot and the Borromean rings are characterised as Hyperbolic knots/links, meaning that their complement can be assigned a metric with constant curvature -1. Thus, they differ fundamentally from the torus knots/links.

To emphasise this key point, we have complemented the figure-eight knot and the Borromean rings with more new knots and links of a somewhat more exotic appearance, taking the form of Turk’s head knots.

** 2. Even the approximation approach on truncating $j_\text{max}$, which did not appear in any previous works by the same authors, is not new. For instance, an analogous truncation approach for realizing arbitrary knots appeared in Bode, Benjamin, and Mark R. Dennis. "Constructing a polynomial whose nodal set is any prescribed knot or link." arXiv preprint arXiv:1612.06328 (2016). **

We thank the referee for raising this relevant point. Even though the approximation techniques may at first sight look similar, we stress that they are used to completely different ends. Bode et. al. are using trigonometric interpolation to in the end construct complex functions whose zeros on the three-sphere represent knots and links. We, on the other hand, use the interpolation to construct a parameterisation of the set of zeros of the function representing the knot in order to calculate knot invariants. Sure, we are both using trigonometric interpolation/Fourier basis expansion for approximating a discrete data set, but our goals differ fundamentally. We include the paper by Bode et. al. in the reference list, and have cited in connection to our parameterisation ansatz. We have furthermore explicitly stated how the two works differ.

Requested Changes

** 1. The authors should consider also citing some other existing/contemporary theoretical and experimental works on physical knots:**

Bode, Benjamin, and Mark R. Dennis. "Constructing a polynomial whose nodal set is any prescribed knot or link." arXiv preprint arXiv:1612.06328 (2016). - please also contrast the approach used in your manuscript with this work.

Sugic, Danica, and Mark R. Dennis. "Singular knot bundle in light." JOSA A 35, no. 12 (2018): 1987-1999.

Li, Linhu, Ching Hua Lee, and Jiangbin Gong. "Boundary states of 4D topological matter: Emergence and full 3D-imaging of nodal Seifert surfaces." arXiv preprint arXiv:1905.07069 (2019).

Larocque, Hugo, Danica Sugic, Dominic Mortimer, Alexander J. Taylor, Robert Fickler, Robert W. Boyd, Mark R. Dennis, and Ebrahim Karimi. "Reconstructing the topology of optical polarization knots." Nature Physics 14, no. 11 (2018): 1079.

**Zhang, Yi. "Cyclotron orbit knot and tunable-field quantum Hall effect." arXiv preprint arXiv:1905.02192 (2019). - which gives knotted analogies to cyclotron orbits **

After considering the suggested additional references, we have come to the following conclusions.

  • Bode et. al. arXiv:1612.06328 (2016) has been cited in connection to our parameterisation ansatz, where we furthermore have highlighted the contrasts with our work.
  • The paper by Li et al, arXiv:1905.07069 (2019) has been mentioned in a Note added, essentially because of the discussion on imaging full Seifert surfaces in systems.
  • Larocque et. al. Nature Physics 14, no. 11 (2018): 1079 has been included in the citation of optical experiments.
  • Sugic et. al. JOSA A 35, no. 12 (2018), 1987-1999 has been included in the citation of optical experiments due to their concrete experiment suggestion.
  • The paper by Zhang has however not be included in the reference list. At first, it looks relevant but is not at the core of the present discussion.

** 2. In the note added pertaining to Ref 50, Ref 50 pertains to Hermitian and not non-Hermitian knots. **

This typo has been fixed, and we thank the referee for pointing this out.

** 3. It will be desirable to provide a more extensive discussion on how the approach used here deviates/complements the approaches of Refs 6 and 22, beyond the level of the relationship between $d_R$ and $d_I$.**

It is our understanding that the original material of our work was not highlighted and clear enough. The key concepts and new insights of this work are twofold.

  • We explicitly construct continuous and discrete models, both Hermitian and non-Hermitian, whose nodal structures attains the form of hyperbolic knots and links, which haven’t been previously discovered.
  • We provide an efficient algorithm for computing an invariant, the Alexander polynomial, of any knotted structure appearing in a generic translational invariant Hamiltonian. In this way, the nodal knots and links can be identified without visual inspection. Furthermore, we show that the polynomial provides higher order Milnor invariants, characterising the Brunnian links.

We have emphasised in a clearer fashion how our work differs and complements earlier works on similar topics in several ways.

  • A new subsection, 2.3, has been added to the manuscript. Here we have included more exotic, and new, hyperbolic nodal structures. In fact, they form the family of $(3,n)$ Turk’s head knots.
  • The originality of computation of knot invariants from a generic Hamiltonian has been stressed in a revised introduction to Section 3.
  • These aspects are mentioned in the last sentence of the Introduction, Section 1.

---

## Round 1 · Referee Report · Anonymous (Referee 2) · 2019-6-24

Strengths

  1. This manuscript provides a pedagogical introduction to knot theories is detailed in the context of nodal line semimetals.

  2. Alongside the pedagogical introduction of the mathematical knot invariants, algorithms for the numerical calculation of the nodal line/knot and its invariants, are provided.

  3. The topics is contemporary and the reported text provides a good entry point for researcher from a wide range of fields.

Weaknesses

  1. It is not clear while reading the paper: what are the novel aspects that are brought forward by the current work?

  2. Can the authors elaborate more on the possible new physics that these structures might entail?

  3. What would protect the different link invariants from having a small perturbation deform between the different knot topologies? What would be the physical signatures of such a transition?

  4. There is not much physics discussed in the work, e.g., the physical implications and signatures of the new nodal structures as well as their robustness to disorder.

Report

In this manuscript, a pedagogical introduction to knot theories is detailed in the context of nodal line semimetals. The two-band spectra of nodal line semimetals are parametrized in a general way for both Hermitian and non-Hermitian systems. This allows for various knot configurations of nodal lines and a general scheme for deriving tight-binding and continuum models that realize these knotted spectra is proposed. Alongside the pedagogical introduction of the mathematical knot invariants that help discriminating between the different knotted spectra, algorithms for the numerical calculation of the nodal line/knot and its invariants, are provided. These algorithms are demonstrated and compared on a number of examples, such as the figure-eight knot and Borromean rings.

Nodal line semimetals are one of the more recent members of the topological materials family. Topology of band structures remains a very interesting field of study with new realizations and implications showing up in a wide range of fields. In this context, providing a pedagogical and hands-on introduction to knot theory is very useful and the authors do a very good job about it. At the same time, I have some reservations before I can recommend publication:

  1. It is not clear while reading the paper: what are the novel aspects that are brought forward by the current work? For a lay reader, it appears that all of the various parts are well-known in various fields. It would be, therefore, useful to have clear statements on what are imported methods from mathematics and computer science and what was newly developed for this work.

  2. Whereas nodal line semimetals can be realized/found nowadays, from the current work I do not see the physical implications/importance of seeking out more complicated nodal link/knot structures. Can the authors elaborate more on the possible new physics that these structures might entail?

  3. One of the main goals of analyzing the topology of different structures, is that there must be an obstruction in moving between the different topologies. This obstruction then manifests with some physical implication, e.g., a topological phase transition between different topological insulators. What would protect the different link invariants from having a small perturbation deform between the different knot topologies? What would be the physical signatures of such a transition?

  4. The nontrivial topology of simple nodal line semimetals can be understood using a bulk winding invariant. Can the authors comments on the generalization of this characterization to the more complex spectral arrangements of nodal knots?

  5. Correspondingly, in simple nodal line semimetals, topological boundary effects appear. What would be the expected boundary modes of the more complex knotted structures? On this note, I think that readers of this more mathematical-type work, would benefit from a short update on the physics state-of-the-art the is devoted to analyzing these effects, see e.g., Phys. Rev. Lett. 121, 166802 (2018) and references therein.

  6. Last, the topology of structures is meaningful only when considering disorder. What would disorder do to the knotted nodal line structures? How robust are the presented algorithms to various disorder distributions?

Requested changes

Properly incorporate answers in the main text to the six point that are detailed in the report.

  • validity: good
  • significance: ok
  • originality: low
  • clarity: high
  • formatting: excellent
  • grammar: excellent

Author:  Marcus Stålhammar  on 2019-07-05  [id 556]

(in reply to Report 2 on 2019-06-24)

Dear Referee,

Thank you for reading our manuscript and providing questions well-suited for interesting discussions and constructive improvements. Below we provide answers, alongside with the corresponding changes to our manuscript. Most importantly, we clarify the originality of our work and now better contrast it with earlier works.

** 1. It is not clear while reading the paper: what are the novel aspects that are brought forward by the current work? For a lay reader, it appears that all of the various parts are well-known in various fields. It would be, therefore, useful to have clear statements on what are imported methods from mathematics and computer science and what was newly developed for this work. **

It has become clear to us when reading the reports that the main findings of our paper are not accessible enough, especially concerning what is genuinely new, which of course originates in the manuscript being highly cross-disciplinary. We stress that it is actually not true that all components were previously known in the previous literature in the various sub-fields. We have clarified this in the following ways.

  • We added a clarifying sentence in the end of the Introduction, Sec. 1, stating what is the main original contributions of this work.
  • We have more clearly spelled out which nodal knots and links that were found earlier and which were newly reported in our work. We also have provided additional new examples of hyperbolic knots and links appearing in these systems, presented in a new subsection, Sec. 2.3.
  • We have clarified that calculations of knot invariants from any given Hamiltonian model was not provided in earlier work. In particular we have clarified how the use of an approximate but topologically equivalent parameterisation of the knots and links in order to do these calculations significantly differ from earlier works. This is added in a revised beginning of Sec. 3, and in the end of Sec. 3.2 and highlights a key new development of the present manuscript.

** 2. Whereas nodal line semimetals can be realized/found nowadays, from the current work I do not see the physical implications/importance of seeking out more complicated nodal link/knot structures. Can the authors elaborate more on the possible new physics that these structures might entail? **

The key new physics is that these new structures have topologically distinct Fermi surfaces. While our discussion mainly focuses on how to observe this in spectroscopic measurements such as ARPES, their phenomenology e.g. in transport is very likely rich and remains an interesting research topic which goes beyond the scope of this work. In this context we also stress that the systems presented in our work are both of the conventional Hermitian type and of dissipative non-Hermitian type and that nodal degeneracies on these systems have quite different physical implications and different Fermi surfaces, even when sharing the exact same nodal structure. While the Hermitian nodal lines provide the Fermi surface, the non-Hermitian nodal lines entail open Fermi surfaces in the form of Seifert surfaces, being a natural higher-dimensional generalisation of Fermi arcs. E.g. in a photonics crystal realisation generalising the 2D experiments in Science 359, 1009 (2018) (Ref. 29) these Fermi-Seifert surfaces would be directly visible in light scattering experiments in direct analogy with the mentioned reference. We have commented on this in the revised manuscript.

Even though the Hermitian knotted semimetals appear similar to nodal line semimetals, the fact that the knotted semimetals require polynomial-type invariants in order to be characterised should be interpreted as if there is something unique about the knotted semimetals. Moreover, this knotting is a feature that is unique in 3D spaces — in 4D and higher every knot can be untied and in 2D projections the knot will give rise to singular (i.e. self intersecting) curves. This means, e.g., that studying 2D features of knotted band structures will not provide full information about the knottedness. This is also discussed below in point 4 and we have provided a discussion of this in a new paragraph in Sec. 4.

However, since the knotted solutions are generic in non-Hermitian systems, their experimental discovery may be more more likely in that realm, e.g. in optical systems or cold atoms. Our discussion when it comes to new physics and possible experimental discovery is therefore mainly focused on this part which we have extended in the new version of the manuscript.

** 3. One of the main goals of analyzing the topology of different structures, is that there must be an obstruction in moving between the different topologies. This obstruction then manifests with some physical implication, e.g., a topological phase transition between different topological insulators. What would protect the different link invariants from having a small perturbation deform between the different knot topologies? What would be the physical signatures of such a transition? **

As a direct physical signature of these transitions, we note that the topology Fermi (Seifert) surfaces of the non-Hermitian systems changes with the transitions. The complexity of the nodal knot or link structure increases, especially when the central circle merges with the previously existing knot — in particular its genus changes. A natural consequence of this is that the minimal genus of the corresponding Fermi surface will increase as well. This feature is something that potentially could be measured in spectroscopical systems, e.g. with light scattering experiments performed in photonic crystals. Accordingly, we have added a paragraph on this in the manuscript at the end of Sec. 2.3.

Also, let us stress that the topologies of the nodal exceptional structures in the non-Hermitian systems are indeed preserved when an arbitrary, but small, perturbation is added to the system. This is because of that 1D nodal structures are generic in non-Hermitian systems. Of course, when this perturbation grows, the topology will eventually undergo some transition. There are several ways to analyse this, and we have provided a specific example (figure-eight knot) of how such transitions may look in Sec. 2.3. These transitions occur generically for all examples of hyperbolic knots and links studied in this work. The perturbation is not added directly to the system, but rather included as a variation of the radius of the three-sphere on which the knot lies. Thus, the knottedness of the band intersection is changed under such perturbations. It should be noted though that this occurs for both (non-fine-tuned) Hermitian and non-Hermitian systems, as long as the perturbations in the Hermitian case preserves the existing symmetry.

** 4. The nontrivial topology of simple nodal line semimetals can be understood using a bulk winding invariant. Can the authors comments on the generalization of this characterization to the more complex spectral arrangements of nodal knots? **

This was indeed one of the main goals of the paper. Nodal unknotted lines are, as said above, characterised by computing an integer bulk winding invariant. When it comes to knots, the structures are so complicated that the usual $\mathbb{Z}_2$ or even $\mathbb{Z}$ invariants are not enough to characterise their topology. In fact, there is to our knowledge no known, or at least computable, unique invariant for knots. As for now, the Alexander, Jones and HOMFLY polynomials are the best known alternatives, and we chose to compute the Alexander polynomial because of its computationally effective algorithms and nice geometrical interpretation (note that we use the Alexander polynomial to in principle extract Milnor invariants of any order). The Alexander polynomial is of course not a $\mathbb{Z}_2$ or $\mathbb{Z}$ invariant, but rather a $\mathbb{Z}[t]$ invariant, that is, an invariant which is an element of the polynomial ring $\mathbb{Z}[t]$. This polynomial then characterises the knottedness of the nodal structure, which is not provided by any integer valued invariant.

To emphasise this question, we have provided a more thorough discussion related to these issues in the beginning of the re-written version of Sec. 3.

** 5. Correspondingly, in simple nodal line semimetals, topological boundary effects appear. What would be the expected boundary modes of the more complex knotted structures? On this note, I think that readers of this more mathematical-type work, would benefit from a short update on the physics state-of-the-art the is devoted to analyzing these effects, see e.g., Phys. Rev. Lett. 121, 166802 (2018) and references therein. **

The 2D-surface states of these knotted structures will have similar boundary states, i.e. the drum-head surface states, as ”usual” nodal line semimetals, see e.g. Phys. Rev. B 96, 201305(R). Thus, in the 2D surface Brillouine zone there seems to be nothing unique separating unknots with certain windings from knotted structures for every projections — since knots only exists in 3D-space, its special properties will be destroyed on a 2D surface. Formally, one would need to scan over all possible 2D-projections in order to be make sure that the obtained surface states really originate from a knotted structure. We have provided a discussion on this in Sec. 4. However, in order to provide similar signatures, it was proposed in an arXiv pre-print arXiv:1905.07069 to consider boundary states of 4D systems, where the boundary states takes the form of Seifert surfaces and hence includes the information related to the knots. We have therefore mentioned the pre-print referred to above in a Note added.

We have also added a reference to Phys. Rev. Lett. 121, 166802 (2018) as suggested.

** 6. Last, the topology of structures is meaningful only when considering disorder. What would disorder do to the knotted nodal line structures? How robust are the presented algorithms to various disorder distributions? **

Generally disorder is a problem in any topological semimetals and standard nodal line degeneracies in 3D as well as the Dirac fermions of graphene in 2D are know to be highly susceptible to disorder. In fact, it by definition ruins the translational symmetry of the system, which in turn ruins at least the simplest available ways of characterising the nodal topology. Nevertheless, some features can survive even quite strong disorder, e.g. the transport properties of Weyl semimetals are known to be robust to finite disorder strengths.

However, if the translational invariance is preserved (or altered such that the unit cell size increases), the story is more straight forward to analyse. The answer to the question is then qualitatively different depending on if Hermitian or non-Hermitian systems are studied. Let us therefore separate the discussion.

Hermitian systems: When it comes to Hermitian systems, the robustness against disorder of the knotted nodal lines does not differ fundamentally from that of regular nodal lines. Thus, there is really nothing new to add to this point.

Non-Hermitian systems: In non-Hermitian band structures, the knotted nodal lines occur generically, in contrast to the Hermitian realm. This is discussed in Phys. Rev. B 99, 161115(R), and the principles are completely analogous in this case. Any small, but arbitrary “disorder” contribution will leave the nodal topology intact. This fact suggests that non-Hermitian systems, such as cold atoms and photonics systems, will be more robust candidates for experimental observations than the knotted semimetals. Studying real translational breaking disorder in these systems would be particularly interesting noting that the generic nature of 3D Weyl fermions make them robust against order and it is an open question if this property carries over also to the generically occurring nodal non-Hermitian degeneracies.

---

## Round 2 · Author Response

The authors are thankful for the obtained reports, contributing with constructive feedback on our first version of the manuscript. We have now adressed all the critique that was raised in the reports, and provided corresponding changes in the updated version of the manuscript.

---

## Round 2 · List of Changes

* Addition of a new subsection, Sec. 2.3, including further more exotic examples of hyperbolic nodal structures, Turk's head knots, and a discussion on transitions between knots with accompanying figures.
* Revised beginning of Sec. 3.
* Added more physical discussions in several sections.
* Treating two additional examples of Brunnian links in Sec. 3.4.
* Highlighting in a clearer manner what is new and how this work differs from and complements other works.
* Correcting various typos.
* Added several references.

---

## Editorial Decision

published